# Imputation Methods for scRNA Sequencing Data

**Mengyuan Wang** [1], **Jiatao Gan** [2], **Changfeng Han** [1], **Yanbing Guo** [2], **Kaihao Chen** [2], **Ya-zhou Shi** [1] **and Ben-gong Zhang** [1,*]

[1] Center of Applied Mathematics & Interdisciplinary Sciences, School of Mathematical and Physical Sciences, Wuhan Textile University, Wuhan 430200, China
[2] School of Computer and Artificial Intelligence, Wuhan Textile University, Wuhan 430200, China
[*] Correspondence: bgzhang@wtu.edu.cn

**Abstract:** More and more researchers use single-cell RNA sequencing (scRNA-seq) technology to characterize the transcriptional map at the single-cell level. They use it to study the heterogeneity of complex tissues, transcriptome dynamics, and the diversity of unknown organisms. However, there are generally lots of technical and biological noises in the scRNA-seq data since the randomness of gene expression patterns. These data are often characterized by high-dimension, sparsity, large number of "dropout" values, and affected by batch effects. A large number of "dropout" values in scRNA-seq data seriously conceal the important relationship between genes and hinder the downstream analysis. Therefore, the imputation of dropout values of scRNA-seq data is particularly important. We classify, analyze and compare the current advanced scRNA-seq data imputation methods from different angles. Through the comparison and analysis of the principle, advantages and disadvantages of the algorithm, it can provide suggestions for the selection of imputation methods for specific problems and diverse data, and have basic research significance for the downstream function analysis of data.

**Keywords:** scRNA sequencing; dropout; imputation; downstream analysis

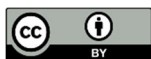

## 1. Introduction

Bulk cell RNA sequencing (RNA-seq) data are usually used in transcriptome analysis, such as transcriptional structure, splicing patterns, genes, as well as transcriptional expression levels [1]. However, in the traditional bulk cell transcriptome sequencing, gene expression is measured by the average reading of a large number of cells, which masks the heterogeneity between cells. Fortunately, scRNA-seq technology provides a more accurate insight into the function of individual cells, and it is also used to reveal the dynamics of heterogeneous cell populations and complex tissues [2–8]. Single-cell transcriptomic analysis has provided unprecedented opportunities to study complex phenomena, such as cancer, tumor, development, and microbial ecology in biological systems [9]. Ting et al. used scRNA-seq technology to obtain high-quality transcripts from mouse pancreatic circulating tumor cells (CTCS), and compared them with scRNA-seq data from human pancreatic cancer patients. They found that the gene SPARC is very much related to the invasion and metastasis of pancreatic cancer [10]. Furthermore, assessing gene expression differences between individual cells has the potential to identify rare populations that are not detected from bulk cell analysis, and the ability to study outlier cells within a population can help people understand the drug resistance in cancer and tumor treatment [11]. In 2015, Kim et al. executed scRNA-seq analysis of cells isolated from a xenograft tumor of a lung adenocarcinoma (LUAD) patient, and they found a subpopulation of tumor cells associated with anticancer drug resistance [12]. Moreover, single cell sequencing can be used to analyze individual cell in tumors and investigate what roles they play, so as to provide new insights into treatment [13]. In 2017, Puram et al. found that the cells exhibiting p-EMT activation were located at the outer area of primary tumors and promote the

invasion and metastasis of tumor cells [14]. Part of the application of scRNA-seq is shown in Figure 1 [15].

In scRNA-seq datasets, more than 50% of the expression in the count matrix is usually found that to be equal to 0 [16,17]. As we know, a lot of technical noise can affect the scRNA-seq data, and one important feature is called "dropout", in which the gene is moderately expressed in one cell but not detected in another [18]. This is because most scRNA-seq techniques usually require heavy amplification due to the picogram level of RNAs in a single cell after reverse transcription. RNAs transcripts may be lost in the reverse transcription and amplification steps, so they will not be detected in subsequent sequencing [6]. The dropout events caused by technology improve the cell-to-cell variability, leading to signal influence on each gene, and obscuration of gene–gene and cell–cell real relationships. Thus, the presence of dropout values will greatly reduce the accuracy of the downstream analysis [19,20]. Moreover, by the selective gene expression, there are many truly unexpressed genes in single cell, which results in a confusion of biological zeros with the technology-induced zeros. Distinguishing between these two cases is a very important but yet not fully resolved problem.

In recent years, there are many imputation methods are proposed to recover gene expression data [21–25]. These methods can be roughly divided into the following categories: probabilistic models are used in some methods to identify the zeros are technique-induced dropout values rather than true zeros, they usually only impute dropout zeros, while not involving other values that are not affected by the dropout events. In this category, we find scImpute [4], SAVER [26], SAVER-X [27], CIDR [28], etc. scImpute estimated dropout rates by using the Gamma-Normal mixture model and population-specific thresholds while SAVER using Possion-Gamma model to pool expression values across genes within each cell [3,15]. SAVER-X performs single-cell analysis by exploiting expression recovery from external data, the approach that combines a Bayesian hierarchical model with a trainable deep autoencoder [27]. CIDR is an ultra-fast algorithm, which estimates the relationship between dropout rate and gene expression level by identifying the neighbors of dropout value [28]. Another class of methods can reconstruct the expression value from the simplified representation of the observed data matrix, including scRMD [6], McImpute [3], etc. scRMD imputes the gene expression value by robust matrix decomposition [6]. McImpute models the gene expression matrix as a low-rank matrix, takes the preprocessed gene expression matrix as the input of the nuclear norm minimization algorithm and recovers the gene expression value of the complete matrix by solving non-convex optimization problems [3]. Some methods adjust the expression values of every cell by utilizing expression levels of "similar" cells. These methods typically change all expression values, containing the dropout values and true zero values as well as the observed non-zero values. For example, the kNN-smoothing uses a Possion distribution and aggregated information from similar cells [29]. With regard to DEWÄKSS denoise expression data using weighted affinity kernels and self-supervised, [30] used a self-supervised technique to tune the parameters. Some authors use scTSSR to impute the "dropout" values [31]. Some methods identify potential spatial representations of cells based on deep learning theory, the observed gene expression matrix is then reconstructed from the estimated potential space. DCA is a neural network-based method that uses deep autoencoding networks for unsupervised learning [32]. DeepImpute and scVI are imputation algorithms based on deep neural network to learn the distribution patterns in the data to accurately impute the dropout values [22,33]. In addition, a recent study proposed a new algorithm called scMOO [34]. These methods recover the gene expression data in various degrees and improve the accuracy of downstream analysis.

Using scRNA-seq data, people can perform downstream analyses, such as cell clustering, differential expression analysis, identification of cell-type specific genes, and reconstruction of differentiation trajectory, which can help people to find rare cell subpopulation within seemingly similar cells. This is particularly advantageous for studying cell-to-cell heterogeneity and cell development dynamics. These downstream analyses are all

highly dependent on the accuracy of the gene expression measurements, so it is imperative to impute the dropouts events caused by technology in the scRNA-seq data by imputation methods [4,6]. However, for massive biological datasets, there is still no universal and effective imputation algorithm to eliminate batch effects and find rare cell types, and there is no clear selection strategy for the imputation method necessary for different characteristics of datasets and specific research problems. Therefore, we classify, analyze, and compare the current advanced single-cell transcriptome sequencing data imputation methods from different perspectives and put forward corresponding recommendations for researchers, it can ensure the accuracy of downstream analysis.

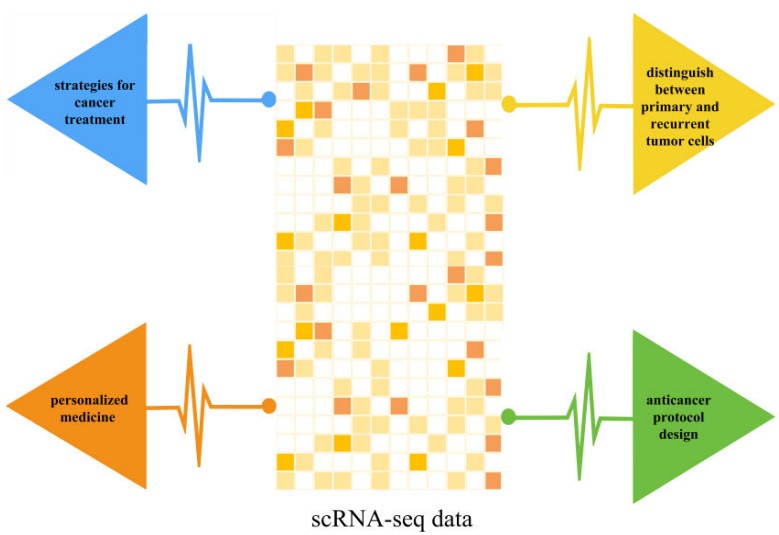

scRNA-seq data

**Figure 1.** Application of scRNA-seq data [15].

## 2. Imputation Methods

For the presence of dropout values of scRNA-seq data, many methods first pre-process and normalize the data, to eliminate the adverse effects of outliers. Then, an important process is to distinguish these zero values is true zero or dropout value caused by technology. Finally, these methods choose a certain strategy to impute the dropout value. As described in [21–25], classifying and comparing currently popular imputation methods is very necessary to help users provide advice when facing different datasets and different needs. Most of the current methods can be divided into the following four cases: (1) Model-based methods: these methods assume the statistical model of technical and biological variability and noise distribution, and impute by estimating the parameters of the distribution. (2) Low-rank matrix-based methods: the method based on the low rank matrix identifies the potential spatial representation of the cell by capturing the linear relationship, and then reconstructs the expression matrix from the low rank that is no longer sparse. (3) Data smoothing methods: the method based on data smoothing typically uses gene expression values in similar cells to adjust all values (usually including zero and non-zero values). (4) Deep learning methods: the method based on deep learning identifies the potential spatial representation of cells based on the deep learning method, and by using estimated potential space to reconstruct the observed expression matrix [21–25]. For detailed division, see Figure 2 [35–77].

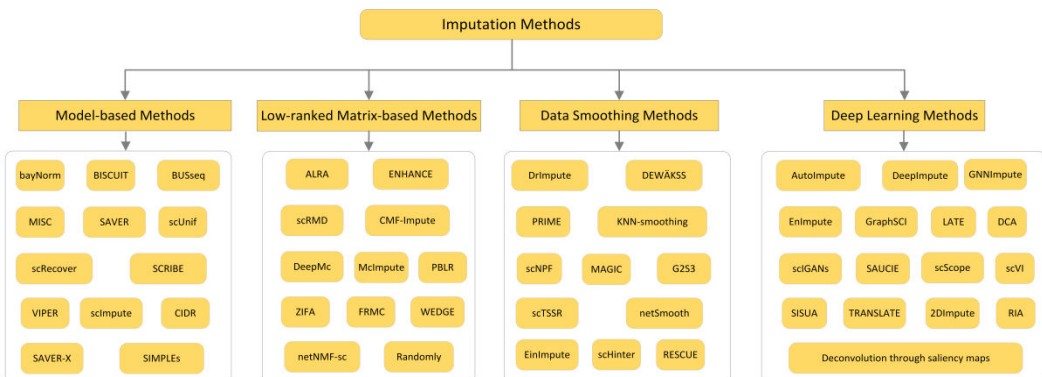

**Figure 2.** Classification method I of imputation method is divided into model-based method, data smoothing-based method, deep learning-based method and low-rank matrix-based method.

The second classification angle divides imputation methods into statistical methods and machine learning methods. Most of the statistical methods make assumptions based on the dataset itself, and then use the original dataset to fill the "dropout" data accordingly. This kind of method does not consider the category of the data object itself, and the filling value is often affected by other types of objects, and the accuracy of the filling result is poor. The common methods include the EM (expectation maximization) filling algorithm, regression analysis, multiple imputation, and so on. Generally speaking, the machine learning method is to classify or cluster the "dropout" dataset and then fill it. Such methods have sprung up with the upsurge of machine learning in recent years. Representative methods are K nearest neighbor filling, K-means filling, Bayesian network, and so on. Among them, the classification method classifies the "dropout" attributes as the target, and then fills them in each category, but when there are too many "dropout" attributes, it is easy to lead to too many categories and low efficiency, meanwhile, the clustering method first clusters the original data to multiple clusters, while according to the similar objects in the cluster to fill, the number of "dropout" attributes will not affect the number of clusters, this kind of method has a wide range of applications, and is also the focus of the current research. The third classification divides imputation methods into (1) global methods and (2) local methods according to the way imputation information is used from observed data, and the fourth classification divides imputation methods into two categories according to whether they are based on Bayesian or not [21].

Using the second method for classification (Table 1) can be divided into statistical methods and machine learning methods. Statistical methods also include mean/mode completer, expectation maximization imputation, hot deck imputation, cold deck imputation, regression imputation. The mean/mode completer include bayNorm. The expectation maximization imputation includes SCRIBE, URSM, SCC [78], JOINT [79], SIMPLEs, ZIFA. The hot deck imputation includes G2S3, scHinter, scNPF, MAGIC, CMF-Impute, deepMc, PBLR, Randomly, WEDGE, FRMC, scRMD and SDImpute. The cold deck imputation include netSmooth, netNMF-sc and ADImpute [80]. The method of regression imputation include CIDR, SAVER, scImpute, VIPER, ZINB-WaVE, MISC, scTSSR, RIA, I-Impute [81] and EinImpute. The main method of machine learning is clustering imputation, include BISCUIT, BUSseq, DEWÄKSS, kNN-smoothing, LSImpute, PRIME, DrImpute, RESCUE, 2DImpute, AutoImpute, SAUCIE, ENHANCE, McImpute, SAVER-X, GraphSCI, SAUCIE, SISUA, TRANSLATE, ZINBAE [82], GNNImpute and ccImpute. The third way can be categorized into local or global methods according to how the imputation information is used from the observed data (Figure 3). The fourth classification (Figure 4) can be divided into Bayesian and non-Bayesian method. For detailed division, see Figures 3 and 4.

**Table 1.** Dropout types, advantage and disadvantages and scope of application of "dropout" values of statistical methods and machine learning methods.

| Method Type | Dropout Type | Advantage | Disadvantage | Scope of Application |
|---|---|---|---|---|
| Mean Completer | Complete random deletion | The operation process is simple | Only the observed information is used, which is subjective, unstable and error | The data scale is small, the missing proportion is small, and the distribution is concentrated |
| Expectation Maximization Imputation | Complete random deletion or random deletion | Good stability and small error | It is not suitable for high-dimensional data | It is applicable to data sets with Normal distribution or approximate Normal distribution |
| Hot Deck Imputation | Random deletion | It has a better effect on maintaining the empirical distribution of variables | The mean square error formula is not clear, and the filling value is easily affected by auxiliary variables | Between data sets collected in the same batch |
| Cold Deck Imputation | Random deletion | The operation process is simple | The filling effect depends on the quality of previous data, and there is estimation deviation | Between data sets collected in different batches |
| Regression Imputation | Random deletion | The operation process is simple and makes full use of the relationship between variables | Without considering the uncertainty of data, it is not suitable for high-dimensional data | It is applicable to data sets with Normal distribution or approximate Normal distribution and multiple auxiliary variables |
| Clustering Imputation | Random deletion | Low variable type requirements, good fitting effect, high stability and small error, suitable for high-dimensional data | The operation process is complex and the time cost is high | It is suitable for any missing pattern and various distribution types of data sets |

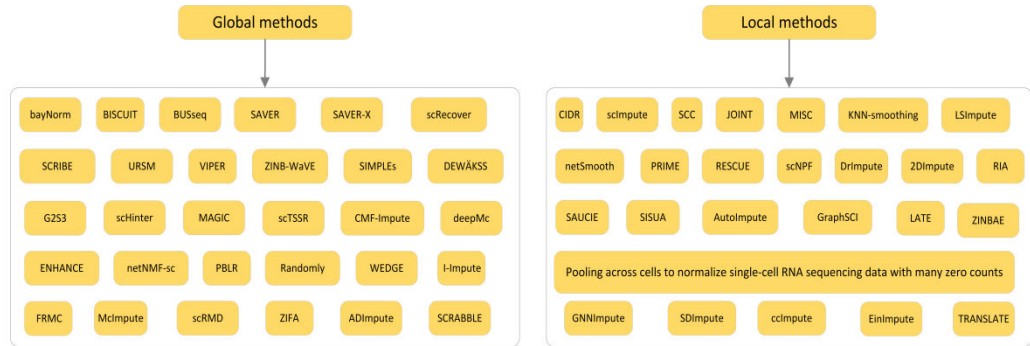

**Figure 3.** The third classification of imputation method for the detailed division of global methods and local methods.

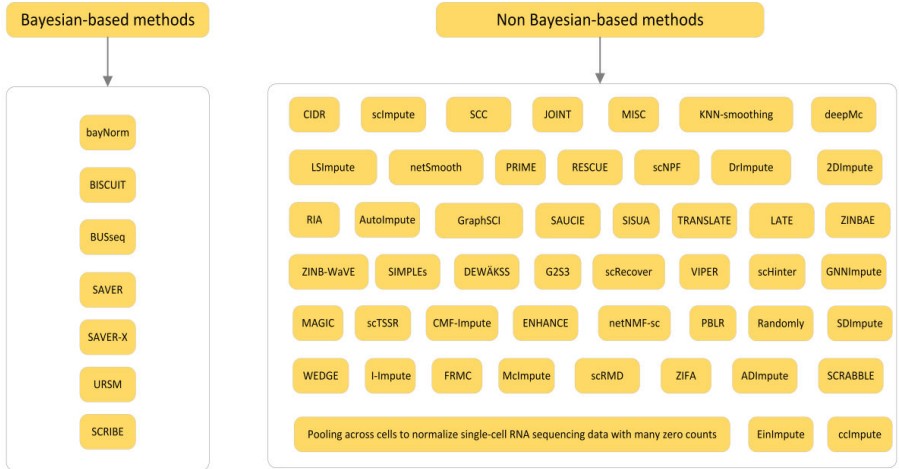

**Figure 4.** The four classification of imputation method for the detailed division of Bayesian and non-Bayesian methods.

Without loss of generality, we chose the first classification method to give detailed description and recommendation. In order to classify and describe the methods clearly, we statement them in much more detail form in the following such as the description in [83].

### 2.1. Model-Based Methods

The model-based method assumes the statistical models of technical and biological variability and noise distribution, and interpolates by estimating the parameters of the distribution, such as scImpute, SAVER, SAVER-X, BISCUIT, VIPER, and bayNorm. The availability links, programming language, whether a package is available, year of presentation, operative system, distribution compatibility and reference of the representative methods based on model see Table 2.

(1) scImpute reduces the dimension of gene expression matrix by principal component analysis (PCA) and uses spectral clustering. The hierarchical model of Gamma distribution and Normal distribution is established for the gene expression in each cell subpopulation, the Gamma distribution can explain "dropout" and the Normal distribution represents the real gene expression level. Then, the Gamma-Normal mixed model is used to estimate which values are affected by dropout, then the dropout values are subsequently imputed by non-negative least square (NNLS) regression using the most similar cells in its neighbors, as shown in Figure 5. The specific workflow is as follows [4]:

(a) The count matrix is normalized according to the library size of each sample (cell), and the logarithmic transformation is performed to prevent the effect of outliers.

(b) Cell sub-population and outliers are detected using spectral clustering.

(c) Dropout values were identified using a Gamma-Normal mixed model.

(d) Information from the same genes is borrowed from similar cells to impute in the dropout values.

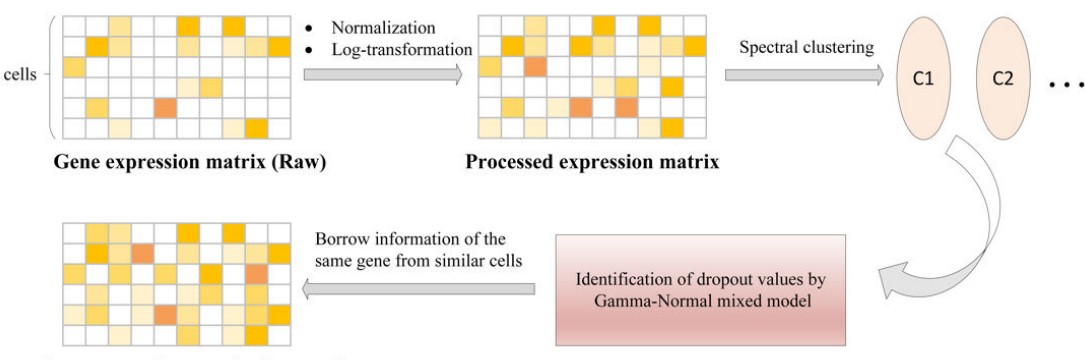

**Figure 5.** The framework of scImpute method [4].

(2) SAVER assumes that the count of each gene in each cell follows the Possion-Gamma mixed distribution. Specifically, the technical noise in the gene expression signal is approximated by the Possion distribution, while the Gamma distribution explains the uncertainty in the real expression. The method does not specify a gamma prior, but instead uses the expression of other genes as the predictors, by using a Poisson Lasso regression to estimate a few prior parameters in the empirical Bayes-like approach. Once the prior parameters are estimated, the SAVER outputs the posterior distribution of the true expression that quantifies the estimated uncertainty, and the posterior mean is used as the expression value for the SAVER recovery, for more detail, please refer to Ref. [26]. We give a simple framework as shown in Figure 6.

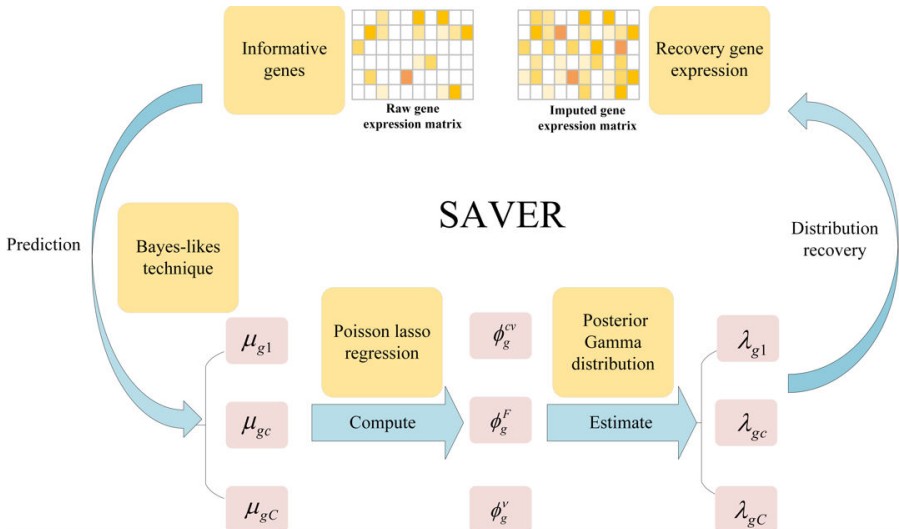

**Figure 6.** The framework of SAVER method [26].

(3) SAVER-X combines a Bayesian hierarchical model with a pre-trained depth autoencoder by recovering the expression of external data for single cell analysis. The pretraining of automatic encoder in SAVER-X includes the sharing network between humans and mice. Specifically, for the target data with UMI counting matrix, SAVER-X trains the target data using an autoencoder without selecting a pre-training model, then uses cross validation to filter unpredictable genes, and uses empirical Bayesian shrinkage to estimate the final denoising value [27].

(4) BISCUIT is a hierarchical Bayesian mixture model with a specific cell scaling, which is realized by incorporating the parameters representing technical variation into the hierarchical Dirichlet process hybrid model, and infers cell clusters according to similar gene expression and determines the technical variation of each cell. In addition, the model can impute dropout values based on cells with similar co-expression patterns [35].

(5) VIPER borrows information from cells with similar expression patterns to impute the expression measurements of interested cells. However, unlike other methods, it does not perform cell clustering before imputation, and it uses a sparse non-generative regression model to actively select a sparse set of local neighborhood cells, the selection of these cells estimates their associated imputation weights in the final estimation step [48].

(6) bayNorm is a Bayesian method for recovering the true counts of scRNA-seq. The likelihood function for the mRNA capture of the proposed method follows a Binomial model, and uses an empirical Bayesian method to estimate its prior from expression values across cells. In order to simulate biological variability, bayNorm makes a priori analysis of the potential real gene expression level by modeling them as a variable following NB distribution. The parameters can then be estimated locally or globally, depending on one's interest in magnifying or not magnifying the differences between cell groups [38].

**Table 2.** The availability links, programming language, whether a package is available, year of presentation, operative system, distribution compatibility and reference of the representative methods based on model.

| Method | Availability | Code | Pkg | Operative System and Distribution Compatibility | Year | Ref |
|---|---|---|---|---|---|---|
| bayNorm | https://bioconductor.org/packages/release/bioc/html/bayNorm.html | R | R | Linux, Mac and Windows | 2020 | [38] |
| BISCUIT | NA | NA | NA | NA | 2016 | [35] |

| BUSseq | https://github.com/songfd2018/BUSseq-Rpackage | C++ | R | Ubuntu 18.04, Mac OX X 10.04 Mojave and Windows 10 Enterprise | 2020 | [41] |
|---|---|---|---|---|---|---|
| CIDR | https://github.com/VCCRI/CIDR | C++/R | R | Linux, Mac and Windows | 2017 | [28] |
| SAVER | https://github.com/mohuangx/SAVER | R | R | Windows 10 and Ubuntu 20.04 LTS | 2018 | [26] |
| SAVER-X | https://github.com/jingshuw/SAVERX | R | R | NA | 2019 | [27] |
| scImpute | https://github.com/Vivianstats/scImpute | R/HTML/Jupyter Notebook | R | NA | 2018 | [4] |
| scRecover | https://github.com/XuegongLab/scRecover | R | R | Unix, Mac and windows | 2019 | [43] |
| SCRIBE | NA | NA | NA | NA | 2019 | [44] |
| URSM | NA | NA | NA | NA | 2018 | [45] |
| VIPER | https://github.com/ChenMengjie/VIPER | C++/R | R | NA | 2018 | [48] |
| SCC | https://github.com/nwpuzhengyan/SCC | R | NULL | NA | 2021 | [78] |
| JOINT | https://github.com/wanglab-georgetown/JOINT | Python | NULL | NA | 2021 | [79] |
| SIMPLEs | https://github.com/JunLiuLab/SIMPLEs | R | NULL | NA | 2020 | [46] |
| MISC | NA | NA | NA | NA | 2018 | [42] |

NA: The research did not mention. NULL: Null value.

### 2.2. Low-Ranked Matrix-Based Methods

The method based on the low rank matrix identifies the potential spatial representation of the cell by capturing the linear relationship, and then reconstructs the expression matrix from the low rank that is no longer sparse. The availability links, programming language, whether a package is available, year of presentation, operative system, distribution compatibility and reference of the representative methods based on low-rank matrix see Table 3.

(1) scRMD imputes the gene expression value by robust matrix decomposition (RMD). It reasonably decomposes the observed gene expression matrix into three contents: potential gene expression matrix, dropouts, and noise, and transforms the dropout value imputation problem into an optimization problem. The optimal gene expression value affected by dropout events is estimated by the alternating direction multiplier method, for more detail, please refer to Ref. [6]. We give a simple frameworkas shown in Figure 7.

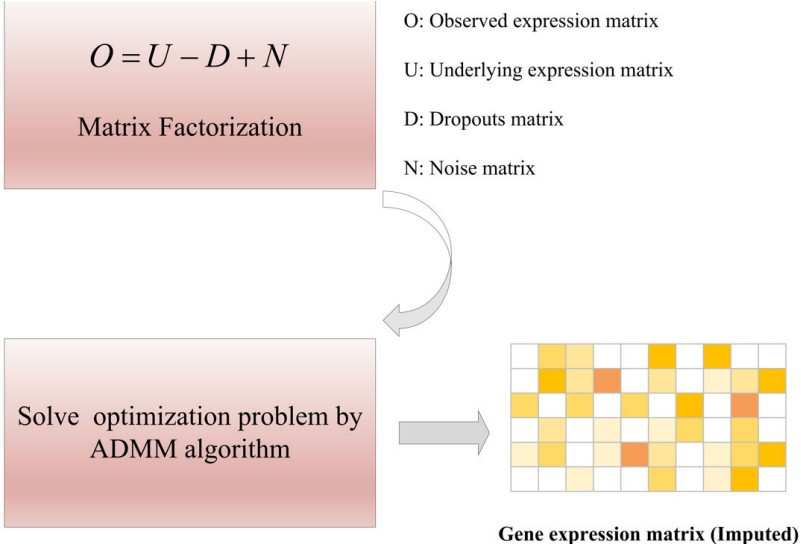

**Figure 7.** The framework of scRMD method [6].

(2) McImpute models the gene expression matrix as a low-rank matrix, takes the preprocessed gene expression matrix as the input of the Nuclear-norm minimization algorithm, and recovers the gene expression value of the complete matrix by solving nonconvex optimization problems. A remarkable feature of McImpute is that it does not

assume that gene expression follows a certain distribution, for more detail, please refer to Ref. [3]. We give a simple framework as shown in Figure 8.

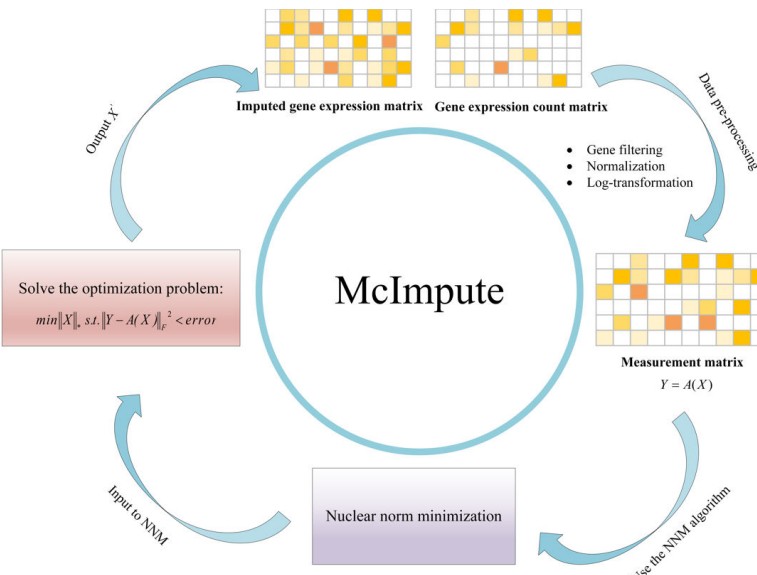

**Figure 8.** The framework of McImpute method [3].

(3) PBLR is a bounded low-rank method based on cell subsets, it not only considers the information of cell heterogeneity and gene expression, but also uses the change of gene expression to impute dropouts. The PBLR first extracts the data for the selected high-variable genes, and calculates the three affinity matrices based on the Pearson, Spearman, and Cosine metrics, respectively. PBLR then learns the consensus matrix by performing SymNMF (symmetric non-negative matrix factorization) on the three affinity matrices INMF (incomplete NMF) of the sub-matrix of the selected genes. PBLR further infers cell sub-populations by performing a hierarchical clustering of the consensus matrix. Finally, PBLR estimates the expression upper bound for the 'dropout' values and recovers the zero gene expression by performing a bounded low-rank recovery model for each sub-matrix determined by each cell sub-population [72].

(4) ENHANCE utilizes PCA and KNN to reduce the noise for the gene expression values. The method consists of two main steps. The first aggregates the expression values to reduce the bias against highly expressed genes. In the second stage, the aggregation matrix is projected onto the first k principal component, where k only represents the real biological difference. Finally, the selected components can obtain the final denoising matrix [69].

**Table 3.** The availability links, programming language, whether a package is available, year of presentation, operative system, distribution compatibility and reference of the representative methods based on low-rank matrix.

| Method | Availability | Code | Pkg | Operative System and Distribution Compatibility | Year | Ref |
|---|---|---|---|---|---|---|
| CMF-Impute | http://bioconductor.org/packages/release/bioc/html/SC3.html | R | R | Linux, Mac and Windows | 2020 | [70] |
| deepMc | https://github.com/hemberg-lab/scRNA.seq.course | TeX/Perl/Dockerfile/R/CSS/Python/Other | NULL | Windows and Unix | 2018 | [71] |
| netNMF-sc | https://github.com/raphael-group/netNMF-sc | Jupyter Notebook/Python | NULL | NA | 2020 | [75] |
| PBLR | https://github.com/amsszlh/PBLR | MATLAB/R/Fortran/C | NULL | NA | 2021 | [72] |
| Randomly | https://github.com/RabadanLab/randomly | Jupyter Notebook/Other | Python | NA | 2020 | [76] |

| WEDGE | https://github.com/QuKunLab/WEDGE | C++/Fortran/CMake/C/Cuda/Python/Other | NULL | Ubuntu20.04 and Windows10, vs2017 and Mac | 2020 | [74] |
|---|---|---|---|---|---|---|
| I-Impute | https://github.com/xikanfeng2/I-Impute | Python/R | R | NA | 2020 | [81] |
| FRMC | https://github.com/HUST-DataMan/FRMC | Python/Jupyter Notebook | NULL | NA | 2021 | [73] |
| McImpute | https://github.com/aanchalMongia/McImpute_scRNAseq | MATLAB | NULL | NA | 2019 | [3] |
| scRMD | https://github.com/XiDsLab/scRMD | R | R | NA | 2020 | [6] |
| ZIFA | https://github.com/epierson9/ZIFA | Python | NULL | Mac | 2015 | [77] |
| ALRA | https://github.com/KlugerLab/ALRA | R | R | OS X, Linux, and Windows | 2018 | [68] |
| EN-HANCE | Python:https://github.com/yanailab/enhance R:https://github.com/yanailab/enhance-R | Python/R | NULL | Linux and Mac | 2019 | [69] |

NA: The research did not mention. NULL: Null value.

### 2.3. Data Smoothing Methods

Based on data smoothing methods typically use gene expression values in similar cells to adjust all values (usually including zero and non-zero values). The availability links, programming language, whether a package is available, year of presentation, operative system, distribution compatibility and reference of the representative methods based on smoothing see Table 4.

(1) MAGIC is a method for explicit and genome-wide inference of single-cell gene expression profiles. This method is based on the concept of thermal diffusion and calculates the dropout gene expression values by sharing information among similar cells. MAGIC constructs the Markov transition matrix by normalizing the similarity matrix of a single cell, and then carries out 'soft' clustering to replace the original expression of genes with their weighted average expression in clustering, thus realizing dropout value imputation, for more detail, please refer to Ref. [36]. We give a simple framework as shown in Figure 9.

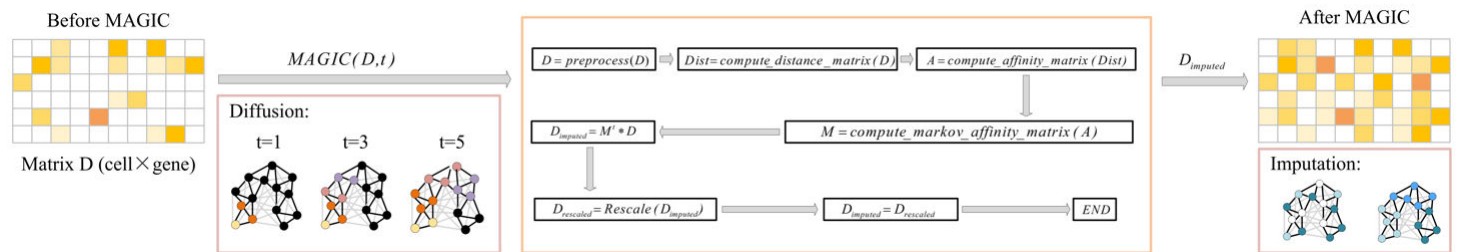

**Figure 9.** The framework of MAGIC method [36].

(2) DrImpute first identifies similar cells based on clustering, and then recovers single cell data by averaging expression values from similar cells. In order to achieve robust estimation, it uses different cell clustering to perform multiple imputing "dropout", then averaging multiple estimation results of the final imputation, as shown in Figure 10. The specific workflow is as follows [49]:

   (a) Raw read counts are normalized by size factor and then log-transformed on the data.
   (b) The first 5% principal components of the similarity matrix are clustered with the k-means using the Pearson and Spearman correlations.
   (c) Borrowing average expression values from similar cells to recover single-cell gene expression data.

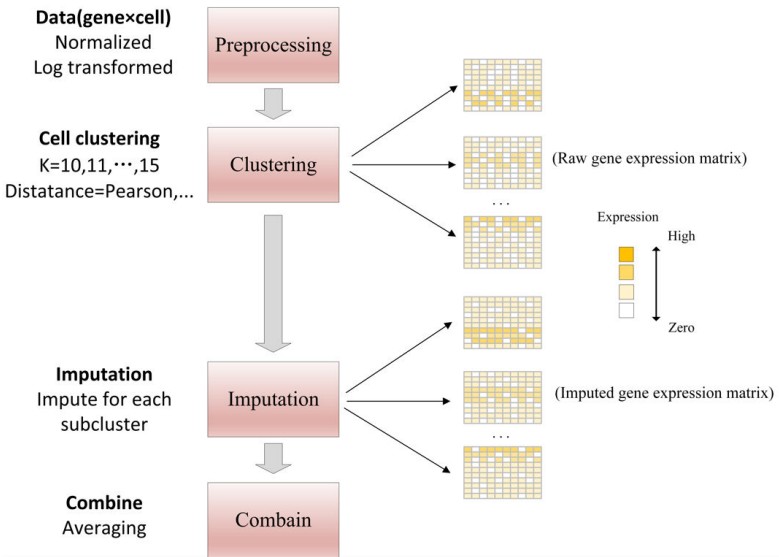

**Figure 10.** The framework of DrImpute method [49].

(3) kNN-smoothing models' technical variance using a Poisson distribution and imputation is conducted via discreet smoothing or variance-stabilization of the expression profiles. kNN-smoothing 2 is a major improvement over the original algorithm, and performs much better whenever the data contain cell populations with very similar expression profiles. kNN-smoothing 2 completely replaces the original version. It takes two parameters (k and d). k is the number of neighbors to use for smoothing (same as in the original version), and d is the number of principal components used for determining the nearest neighbors in each smoothing step [29]. MAGIC, DrImpute, and kNN-smoothing are three classical data smoothing methods.

**Table 4.** The availability links, programming language, whether a package is available, year of presentation, operative system, distribution compatibility and reference of the representative methods based on smoothing.

| Method | Availability | Code | Pkg | Operative System and Distribution Compatibility | Year | Ref |
|---|---|---|---|---|---|---|
| DEWÄKSS | https://gitlab.com/Xparx/dewakss/-/tree/Tjarnberg2020branch | Python | Python | NA | 2021 | [30] |
| G2S3 | https://github.com/zwang-lab/g2s3 | MATLA/R | NULL | NA | 2021 | [55] |
| kNN-smoothing | https://github.com/yanailab/knn-smoothing | Python/R/MATLAB | NULL | NA | 2018 | [29] |
| netSmooth | https://github.com/BIMSBbioinfo/netSmooth | R | R | Linux, Mac and Windows | 2021 | [37] |
| PRIME | https://github.com/hyundoo/PRIME | R/C++ | R | NA | 2020 | [52] |
| RESCUE | https://github.com/seasamgo/rescue | R/Python | R | NA | 2019 | [53] |
| scHinter | https://github.com/BMILAB/scHinter | MATLAB | NULL | NA | 2019 | [50] |
| scNPF | https://github.com/BMILAB/scNPF | R | R | NA | 2019 | [54] |
| MAGIC | https://github.com/DpeerLab/magic | Jupyter Notebook/Python/MATLAB | Python\|R | NA | 2018 | [36] |
| scTSSR | https://github.com/Zhangxf-ccnu/scTSSR | R | R | NA | 2020 | [31] |
| DrImpute | https://github.com/gongx030/DrImpute | R/C++ | R | Linux | 2018 | [49] |

NA: The research did not mention. NULL: Null value.

### 2.4. Deep Learning Methods

This type of approach identifies the potential spatial representation of cells based on the deep learning method (capturing nonlinear relationships) and then reconstructs the observed expression matrix from the estimated potential space. The availability links,

programming language, whether a package is available, year of presentation, operative system, distribution compatibility and reference of the representative methods based on deep learning see Table 5.

(1) DCA is an imputation method based on automatic encoder, it uses a negative binomial noise model with or without zero expansion. By considering the count distribution, super-dispersion, and sparsity of data, it can capture nonlinear gene–gene dependence. DCA learns gene-specific distribution parameters by minimizing the reconstruction error in an unsupervised manner, rather than reconstructing the input data itself, for more detail, please refer to Ref. [32]. We give a simple framework as shown in Figure 11 (a).

(2) LATE uses the initial values of a randomly generated parameter to train an autoencoder on highly sparse scRNA-seq data, and the TRANSLATE method builds on LATE, further using the reference gene expression data set to provide LATE with an initial set of parameter estimates. Hence, the user can train the autoencoder on a reference gene expression dataset and then use the weights and biases as initial values for imputing the dataset of interest. These algorithms are highly scalable in graphics processing units and can handle more than 1 million cells in a few hours [60].

(3) scVI is a fully probabilistic method for the standardization and analysis of scRNA-seq data. The method is based on a hierarchical Bayesian model with conditional distribution specified by a deep neural network. The transcriptome of each cell is encoded into low-dimensional potential vectors of normal random variables by nonlinear transformation. This potential representation is decoded by another nonlinear transformation to generate an a posteriori estimate of the distributional parameters of each gene in each cell, for more detail, please refer to Ref. [84]. We give a simple framework as shown in Figure 11 (b). Both the deep learning method and the low-rank matrix representation method use the idea of data reconstruction.

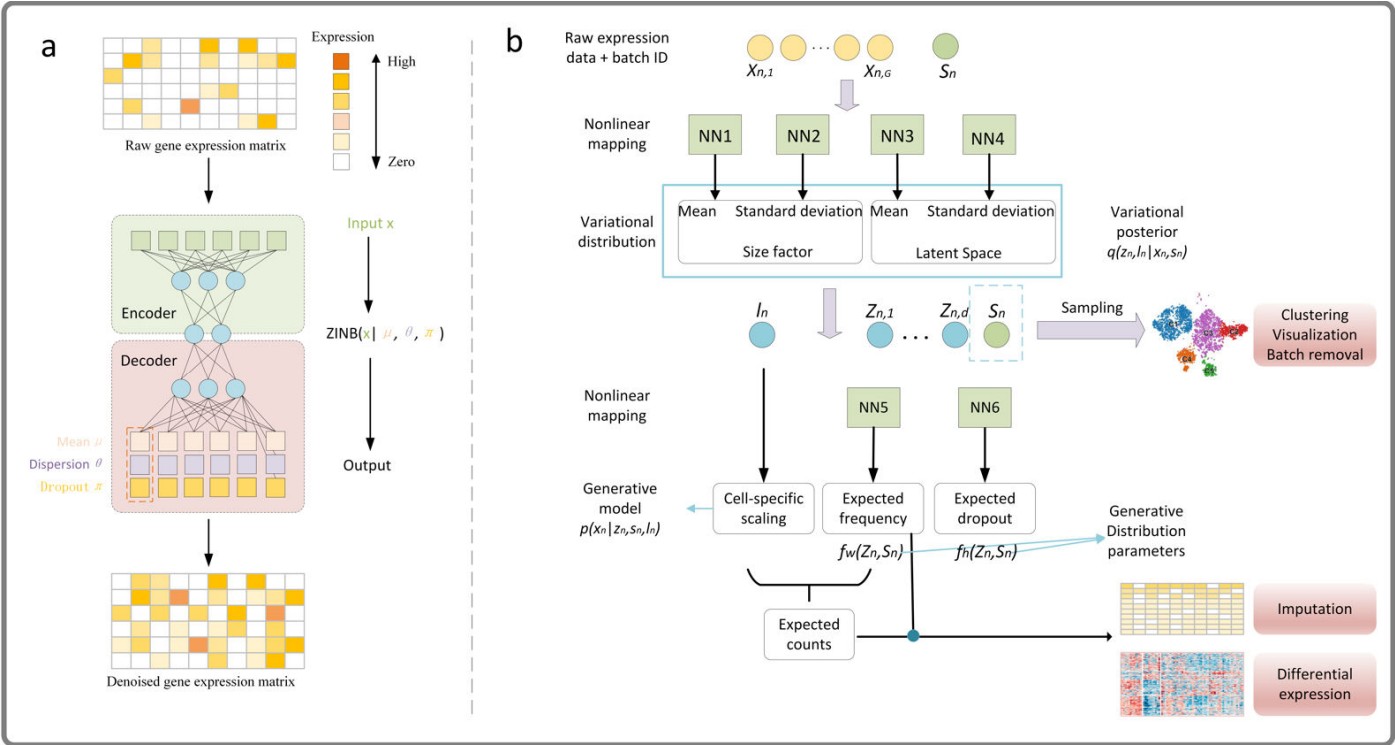

**Figure 11.** The framework of DCA (**a**) and scVI (**b**) method [32,84].

**Table 5.** The availability links, programming language, whether a package is available, year of presentation, operative system, distribution compatibility and reference of the representative methods based on deep learning.

| Method | Availability | Code | Pkg | Operative System and Distribution Compatibility | Year | Ref |
|---|---|---|---|---|---|---|
| 2DImpute | https://github.com/zky0708/2DImpute | R | R | NA | 2020 | [65] |
| AutoImpute | https://github.com/divyanshu-talwar/AutoImpute | Python/R | NULL | NA | 2018 | [57] |
| GraphSCI | https://github.com/biomed-AI/GraphSCI | Jupyter Notebook/Python | NULL | NA | 2021 | [59] |
| RIA | NA | NA | NA | NA | 2019 | [66] |
| SAUCIE | https://github.com/KrishnaswamyLab/SAUCIE/ | Python | NULL | NA | 2019 | [62] |
| SISUA | https://github.com/trungnt13/sisua | Jupyter Notebook/Python/HTML | Python | NA | 2020 | [64] |
| TRANSLATE | https://github.com/audreyqyfu/LATE | Python/Jupyter Notebook/Shell/R | Python | NA | 2020 | [60] |
| ZINBAE | https://github.com/ttgump/ZINBAE | Python | NULL | NA | 2021 | [82] |
| DCA | https://github.com/theislab/dca | Python | Python | NA | 2019 | [32] |
| scVI | https://github.com/YosefLab/scVI | Python | Python | NA | 2018 | [84] |
| scIGANs | https://github.com/xuyungang/scIGANs | Python/shell/R | NULL | Linux/Unix-based systems | 2020 | [61] |
| DeepImpute | https://github.com/lanagarmire/DeepImpute | Python/Makefile/Dockerfile | Python | NA | 2019 | [33] |
| scScope | https://github.com/AltschulerWuLab/scScope | Python | Python | Ubuntu 14.4 | 2019 | [63] |
| EnImpute | https://github.com/Zhangxf-ccnu/EnImpute | R | R | NA | 2019 | [58] |

NA: The research did not mention. NULL: Null value.

## 3. Conclusions and Discussion

We review most imputation methods used in recent references. And we find that these methods can be divided into four categories. The first category includes imputation methods that use probability models to directly represent sparsity. In the imputation function, it may not be possible to distinguish between biological zeros and technical zeros, usually only technical zeros are considered. This method produces fewer false positives, but it depends on the homogeneity or heterogeneity of the data. The second category includes methods for smoothing or adjusting zero and non-zero values through average expression values or their diffusion. This method helps to reduce noise, but it may produce many false positives. Interestingly, in datasets with small effect size genes, the first kind of method may be better than the second kind of algorithm. The third category is based on low-rank matrix, and the fourth category is based on deep learning neural network. The third and fourth methods are based on data reconstruction. The method based on low-rank matrix captures the linear relationship, while the deep learning method deals with the nonlinear relationship.

Through the analysis of various imputation methods, we find that CIDR, MISC, PRIME, RESCUE, RIA, TRANSLATE, and others provide the best performance in terms of computing time. In terms of memory consumption, SCC, G2S3, and others provide the highest memory efficiency. In terms of scalability, AutoImpute, GraphSCI, netSmooth, SAUCIE, scImpute, TRANSLATE, MAGIC, kNN-smoothing etc. demonstrate high scalability as the number of cells in the dataset increases. Based on a comprehensive comparison of time, memory, and scalability, ENHANCE, MAGIC, SAVER, SAVER-X, and scVI provide the best overall performance.

We note that most methods seem to be difficulty to deal with non-UMI datasets, especially the full-length scRNA-seq datasets. This is because comparing with the large number of variables such as genes, the observations such as cells are always small. At this point, although the bayNorm is originally designed for the scRNA-seq protocol that

includes UMI, it also applies to non-UMI data. In addition, we also note that methods using biological variability and technical noise assumptions (SAVER, SAVER-X, scVI, etc.) usually show better performance, which is because they use a priori knowledge in their algorithms. It is also shown that model-based methods usually perform well in imputation and expression recovery, but they are usually costly and poor in the enhancement of cell similarity. Techniques based on matrix theory show good performance in characterizing cell similarity, as well as noteworthy scalability, even in the case of big datasets. At last, data smoothing methods usually show good performance, but there are significant differences according to specific tasks.

We believe that the best imputation method depends on genes and datasets. That is, there is no single best way to perform. For example, a network-based approach may perform well in cell types that meet the co-expression model assumptions, while it may fail (for the same gene) in other cell types that do not meet these assumptions. Therefore, the best imputation method depends on genes and datasets. We should no longer look for a single best imputation method. Instead, the future task will be to find the best way for a specific combination of genes and experimental conditions.

Finally, we point out that imputation methods for precision-imputing of scRNA-seq data in the presence of dropout values pave the way for specific downstream analyses, for example, where improved reliability and accuracy of scRNA-seq data enable more accurate investigation of single-cell genotypes and phenotypes. Moreover, this allows one to better estimate metabolic fluxes from scRNA-seq data, and act into cancer metabolic studies. In the face of the emergence of massive data, the recommended selection of imputation methods from different angles enhances the accuracy and reliability of the data, and has basic research significance for the further development of the concept of model-driven and data-driven combination. It provides data support for the in-depth study and experimental verification of the essence and cell fate determination mechanism of systems biology and molecular biology. Further, since human cancer is a complex ecosystem of cells with different molecular characteristics, this intra-tumoral heterogeneity poses a major challenge to the diagnosis and treatment of cancer, and recent advances in single-cell technologies such as scRNA-seq have brought unprecedented insights into cellular heterogeneity, from which people can discover the metabolism of different cell types, thus enabling early diagnosis of cancer [85,86]. Therefore, the recovery of gene expression value of single-cell transcriptome sequencing data is of great significance for overcoming major disease problems and the development of precision medicine and smart medicine in the future.

**Author Contributions:** Writing—original draft, M.W. and J.G.; writing—review & editing, M.W., Y.S. and B.Z.; visualization, C.H., K.C. and Y.G.; supervision, B.Z.; project administration, B.Z.; funding acquisition, B.Z. All authors have read and agreed to the published version of the manuscript.

**Funding:** This work is supported by the National Natural Science Foundation of China (NSFC) under Grant No. 11971367.

**Institutional Review Board Statement:** Not applicable.

**Informed Consent Statement:** Not applicable.

**Data Availability Statement:** Not applicable.

**Conflicts of Interest:** The authors declare no conflict of interest.

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
