# Peer review of "Imputation Methods for scRNA Sequencing Data"

_applsci, doi:10.3390/app122010684_

Round 1
Reviewer 1 Report
This is a very interesting manuscript about scRNA-seq data imputation methods. The authors not only classify the methods but also analyze and compare different ones, from different perspectives. Currently, there is hype on scRNA-seq, which means that more and more methods and technologies are being developed on a weekly basis. For this reason, it is important to have manuscripts that propose the classification and comparison of new methods, as Wang et al described here.
The only thing I noticed was the lack of references for some phrases, especially in the introduction. I suggest reviewing that.
Author Response
Dear reviewer,
Thank you so much for your time to carefully check our manuscript. Your comments and suggestions are exactly helpful and valuable for us. They make our manuscript much more smoothly. For detail revising,please see the attachment.
Sincerely yours,

Reviewer 2 Report
The review is useful and extensive into describing the challenges and approaches to process data from single-cell RNA seq projects. The imputation methods are described in detail. I consider an excellent decision to mention the programs, the actual places where they can be downloaded and their use.
From the point of view of the user, and from an interdisciplinary audience of the journal, I suggest that practical details on the operative system and distribution compatibility (RedHat, OpenLinux, Mac OSx?) could be mentioned. Also, a descriptive protocol of processing data would be useful for people starting in the field.
The technical and statistical description of the methods is terse, and also, in my opinion, too technical to the reader from the biological coming into bioinformatics. An effort into providing more examples, cartoons, or more descriptive text would be appreciated.
As an example of a more hands-on style, sort of the "Methods in Enzymology" would be much helpful, where the workflow is easily described.
https://www.sciencedirect.com/science/article/abs/pii/S0076687920302482
Author Response
Dear reviewer,
Thank you so much for your time to carefully check our manuscript. Your comments and suggestions are exactly helpful and valuable for us. They make our manuscript much more smoothly. For detail revising, please find them in the attachment.
Sincerely yours,

Reviewer 3 Report
This manuscript offers an interesting and robust viewpoint and it can be accepted in the present form.
Author Response
Dear reviewer,
Thank you so much for your time to carefully check our manuscript. Thank you very much for your kind comments.
Best wishes,
Sincerely Yours,
All authors
